# Nanogel Containing Gamma-Oryzanol-Loaded Nanostructured Lipid Carriers and TiO_2_/MBBT: A Synergistic Nanotechnological Approach of Potent Natural Antioxidants and Nanosized UV Filters for Skin Protection

**DOI:** 10.3390/ph16050670

**Published:** 2023-04-28

**Authors:** Omolbanin Badalkhani, Patrícia C. Pires, Maryam Mohammadi, Soraya Babaie, Ana Cláudia Paiva-Santos, Hamed Hamishehkar

**Affiliations:** 1Research Center for Pharmaceutical Nanotechnology, Tabriz University of Medical Sciences, Tabriz 5165665811, Iran; 2Department of Pharmaceutical Technology, Faculty of Pharmacy of the University of Coimbra, University of Coimbra, 3000-548 Coimbra, Portugal; 3REQUIMTE/LAQV, Group of Pharmaceutical Technology, Faculty of Pharmacy of the University of Coimbra, University of Coimbra, 3000-548 Coimbra, Portugal; 4Health Sciences Research Centre (CICS-UBI), University of Beira Interior, Av. Infante D. Henrique, 6200-506 Covilhã, Portugal; 5Department of Food Science and Engineering, Faculty of Agriculture, University of Kurdistan, Sanandaj 6617715175, Iran; 6Physical Medicine and Rehabilitation Research Center, Tabriz University of Medical Sciences, Tabriz 5166614756, Iran; 7Drug Applied Research Center, Tabriz University of Medical Sciences, Tabriz 5165665811, Iran

**Keywords:** antioxidant, cosmeceuticals, nanogel, nanostructured lipid carriers, natural-based, skin protection

## Abstract

The human skin is a recurring target of external aggressions, such as UV radiation, leading to exacerbation of the aging process and the occurrence of skin diseases, such as cancer. Hence, preventive measures should be taken to protect it against these aggressions, consequently decreasing the chance of disease development. In the present study, a topical xanthan gum nanogel containing gamma-oryzanol-loaded nanostructured lipid carriers (NLCs) and nanosized UV filters TiO_2_ and methylene bis-benzotriazolyl tetramethylbutylphenol (MBBT) was developed to assess their synergistic potential in having multifunctional skin beneficial properties. The developed NLCs contained the natural-based solid lipids shea butter and beeswax, liquid lipid carrot seed oil, and the potent antioxidant gamma-oryzanol, with an optimum particle size for topical application (<150 nm), good homogeneity (PDI = 0.216), high zeta potential (−34.9 mV), suitable pH value (6), good physical stability, high encapsulation efficiency (90%), and controlled release. The final formulation, a nanogel containing the developed NLCs and the nano UV filters, showed high long-term storage stability and high photoprotection ability (SPF = 34) and resulted in no skin irritation or sensitization (rat model). Hence, the developed formulation showed good skin protection and compatibility, demonstrating promise as a new platform for the future generation of natural-based cosmeceuticals.

## 1. Introduction

Human skin vulnerability to the harmful effects of external aggressions is an extremely important factor in exacerbating the aging process and increasing the chance of the occurrence of common skin diseases. The most impactful of these aggressions is UV radiation from sun rays. Different wavelengths will cause different damaging effects, with UVB rays, which have wavelengths between 280 and 320 nm, being prone to causing skin inflammation, burns, and redness, mainly on the outer surface layer of the skin, and UVA rays, which can be categorized as UVA1 (wavelengths between 340 and 400 nm) and UVA2 (wavelengths between 320 and 340 nm), being prone to causing skin blemishes, freckles, sagging, the destruction and weakening of collagen, DNA damage, and wrinkles due to being able to penetrate deeper into the skin. Furthermore, natural defense mechanisms alone are not sufficient to protect the skin and its cells against such harmful effects of UV radiation. Hence, the inflammatory and oxidative effects of UV radiation on human skin will lead to an increased propensity for the development of several skin diseases, such as atopic dermatitis, psoriasis, acne, and even skin cancer, and are also the main cause of premature/accelerated aging and the development of a decreased capacity for skin regeneration (Figure 1) [1,2,3,4,5,6,7,8].

Therefore, preventive measures, such as using topical preparations with appropriate protection capability against UV radiation, and antioxidant and/or anti-inflammatory effects, are necessary [9,10]. Nevertheless, topical drug delivery is a quite challenging task, as enabling active substances to reach even the most superficial layers is quite difficult, as the outermost layer of the epidermis (stratum corneum) acts as a natural protective barrier, hindering active substance and drug penetration. To overcome the skin penetration problems of topical products, various techniques have been evaluated so far, with the purpose of breaking down intercellular lipids and allowing the molecules to penetrate the drug into the skin [11,12,13].

Amongst the several techniques that have been studied over the years, nanostructured lipid carriers (NLCs) have proven to be very effective in increasing skin permeability and the effectiveness of drugs and other active substances. NLCs have a small particle size, which leads to a high contact surface being available to interact with the biological membranes and have the capability of protecting the encapsulated molecules against chemical and metabolic degradation (either originating from environmental factors or related to postadministration effects), which is particularly important for sensitive active ingredients. Additionally, they have been reported to increase a formulation’s physical and chemical stability, have high drug loading for hydrophobic molecules (including oils), lead to controlled drug release, and increase drug permeability through biological membranes. All these factors, added to these systems’ biodegradability, biocompatibility, nontoxicity, and low irritability, which come from the fact that no toxic organic solvents are used in their composition, make NLCs great candidates for drug delivery. Additionally, they are reported to have high reproducibility, easy scale-up, and low production process costs, making them one of the most attractive and marketable choices for topical drug delivery. Since 2005, a good number of cosmetic products containing NLCs have entered the global market. NLC systems are attractive technologies in relation to industrial needs, namely scale-up, qualification and validation, simple technology, low cost, tolerability, etc. The first products to reach the market were Nanorepair and Nanovital by Dr. Rimpler GmbH in Germany in 2005, followed in 2006 by three products in the line IOPE by AmorePacific in South Korea. Therefore, nowadays, many cosmetic products containing lipid nanocarriers are available in the market, which are free of any risks or side effects for skin protection [14,15,16,17].

In the case of sunscreen products, some of them comprise incompatible chemical UV filters, such as oxybenzone, avobenzone, octocrylene, and octinxat, which in long-term use can cause many complications, such as allergies, cell damage, hormonal disorders, and defects in the normal function of tissues and internal organs of the consumer. Some others involve titanium dioxide (TiO_2_) as a physical UV filter, but due to its usually large particle size, this filter has a tendency to create a white chalky layer on the skin surface, a fact that could be considered non-ideal by consumers [18,19]. Nevertheless, if this type of filter is used at the nanoscale, more particles are placed in the path of UV light, and these substances become transparent, thus improving consumer opinion and enabling their use in sunscreen products [20,21,22,23,24,25].

However, there has been increasing concern about the concentration of synthetic UV filters in formulations, as they have reported toxicity in human endocrine, urinary, and reproductive systems and due to environmental reasons, as they will also end up being harmful to marine organisms [26,27,28]. However, the concentration of these filters in formulations can be decreased with the addition to the preparation of antioxidant substances, as they can act synergistically in protecting the skin against the harmful effects of UV radiation. Additionally, antioxidant substances will also have a stabilizing effect in formulations, which adds to their value. Gamma-oryzanol (GO) is a natural antioxidant of plant origin that consists of a mixture of at least ten phytosterol ferulates [29]. GO is considered to have an antioxidant effect that is up to 10 times higher than vitamin E in rice bran [27]. Additionally, it not only has a potential skin-protecting effect due to its high antioxidant abilities but also absorbs UV light, has intrinsic sun protection factor (SPF)-boosting properties, and has been reported to exhibit antiaging effects as well. In spite of these merits, GO has poor skin absorption due to having a high molecular weight (>500 Da) and is also highly lipophilic, which makes it hard to solubilize at high strength without using potentially toxic solvents. These characteristics provide extra challenges in its formulation, and consequently, GO application in topical products has been limited.

A strategy to overcome the issues presented by GO can be the encapsulation of this compound in NLCs using environmentally friendly liquid vegetable oils and natural solid lipids to create a safe and effective carrier for its topical application [30,31,32]. Among the many natural-derived oils that can be chosen from, carrot seed oil has proved to be a good candidate for skin protection and disease prevention, as it has been proven to have not only antibacterial and antifungal activity but also antiaging potential due to its antioxidant activity (1-diphenyl-2-picrylhydrazyl and nitric oxide free radical scavenging) and SPF and has proven skin compatibility as well (no irritation) [33,34,35,36]. Another promising and natural-sourced oil in shea butter has proven to have several relevant activities as well, such as antibacterial, antifungal, wound and burn healing, and anti-inflammatory, antioxidant, and even chemopreventive action [37,38,39]. As for solid lipids, beeswax is proven to be a quite advantageous choice, as it has reported antibacterial, antiviral, antiparasitic, anti-inflammatory, antioxidant, wound-healing, and anticancer properties [40,41,42].

Additionally, in order to safely reduce the photocatalytic activity of nanoparticles in living tissues, their surface can be covered with various environmentally friendly materials, especially in aqueous dispersion forms, and confer additional beneficial and efficacious photoprotection [43]. Therefore, a combination of two European Union-authorized inorganic and organic UV filters has been used and involves an aqueous dispersion of coated TiO_2_ nanoparticles in rutile crystal form and a viscous dispersion of colloidal nanoparticles of methylene bis-benzotriazolyl tetramethylbutylphenol (Nano-MBBT). Due to the deposition of these nanoparticles on the skin, their permeability to the inner layers of the skin and the resulting safety hazards are reduced. Moreover, because of the synergistic effects of the applied UV filters, a safe formulation with appropriate coverage and higher protective performance can be created [44,45,46,47,48].

Hence, the purpose of this work was to create a natural-based multifunctional product, namely a topical xanthan gum nanogel containing GO-loaded NLCs and nanosized UV filters TiO_2_ and MBBT (Figure 2) for skin protection against UV rays with multiple beneficial properties, such as potent antioxidant activity and a reduction in skin disease manifestation.

## 2. Results and Discussion

### 2.1. GO-NLC Formulation Development, Characterization, and Stability

As shown in Table 1, the optimum blank (without GO) NLC formulation composition was assessed using several varying parameters, including the concentration and type of hydrophilic surfactant, which was either Tween^®^ 80 or Poloxamer 407; the concentration and type of hydrophobic cosurfactant, either PGE or PGPR; and the oil–lipid total content, varying from 7.6 to 9%. Considering the well-known role of the type and ratio of liquid oils, solid lipids, and surfactants in the structure of NLCs, our main focus was on developing an effective NLC-based sunscreen formulation and partly replacing UV filters by including and maximizing the amount of the potent plant antioxidant GO in the formulations. Additionally, we chose to use natural oil–lipids due to their proven skin protection effects and environmentally friendly nature, and these components allowed a smaller amount of UV filters in the formulation to be used, with synergistic effects on the SPF value, and led to a suitable texture in terms of consistency and sensorial experience on the consumer’s skin. The technological formulation characterization parameters and skin protection efficacy of the developed formulations were assessed and are discussed in the following sections.

The optimum blank NLC formulation was found to be F15, made of 1.8% PGE, 2.4% Tween^®^ 80, and 9% oil–lipid (Table 1). This formulation led to a smaller particle size (148 nm) and narrower size distribution (0.21 PDI value) than all other tested variations (Table 1 and Figure 3A). Additionally, the obtained particle size was within the optimum values for topical application, which is between 100 and 200 nm, as particles with smaller sizes will tend to permeate the skin and reach the systemic circulation (which is not intended in the case of topical delivery), and particles with bigger sizes will tend to be easily removed from the application site. Moreover, the PDI value of the selected formulation was around 0.2, which shows great homogeneity, representing a good potential for high stability [49,50,51,52,53].

Furthermore, the particle size and PDI obtained from F16, respectively, 146 nm and 0.20 (Table 1 and Figure 3E), showed that drug incorporation into the NLCs did not significantly change these formulation characterization parameters. This finding shows that the encapsulation of GO into the NLCs did not lead to larger particle size, instability, or decrease in the uniformity of the developed nanoparticles, and hence, the optimized nanosystem maintained its good properties.

The SEM (Figure 4A) and TEM (Figure 4B) micrographs of the synthesized GO-loaded NLC dispersions revealed a uniform amorphous spherical structure, without free GO crystals, within the nanometer range, confirming the DLS analysis results. Additionally, the GO-loaded NLC formulation showed high encapsulation efficiency (90%), which confirmed that the drug loading was indeed successful.

As for the particle surface charge, the optimized blank NLCs led to an electronegative and high zeta potential value, with a value of −29.2 mV (Figure 3B). This characteristic could contribute to the increase of a formulation’s physical stability, as the particles will repel each other and hence have a lower tendency for the occurrence of instability phenomena, such as aggregation and sedimentation. Furthermore, drug incorporation into the NLCs did not significantly change their zeta potential values, in comparison with blank NLCs (−34.9 mV, Figure 3F), and hence, they maintained their potential for stabilization due to surface charge repelling forces.

In terms of short-term real-time stability, after 1 month from production, no significant change was found in the NLC particle size (149 nm), PDI (0.25), or zeta potential (−26.5 mV) (Table 1 and Figure 3C,D), which reflects good physical stability of the NLC dispersion. Furthermore, the expected organoleptic stability of the prepared nanogel was assessed by resorting to accelerated stability tests, which were performed at different storage conditions (8 °C, 25 °C, 40 °C, and 40 °C + 75% RH for 6 months). As there was no visual observation of phase separation, no significant change in pH values, and no change in the color or odor of the formulation at predetermined times (months 0, 1, 2, 3, and 6), it was concluded that the formulated nanogel had adequate long-term stability as well.

The in vitro drug release studies were meant to assess the extent and speed at which GO was released from within the developed NLCs. The results showed that the formulations showed a controlled drug release (Figure 5), reaching around 80% release after 72 h. These results are in agreement with previous studies, which indicate that NLCs provide a controlled drug release pattern [54]. Sustained drug release will allow drug levels to maintain constant for a long period of time, which is ideal for prolonged effects. Furthermore, there was no initial burst release, which is a further indication of formulation stability. Furthermore, the best model for the release kinetics proved to be the RPT model, having a maximum value of R^2^ (=0.994) and a minimum amount of mean error percentage (=11.78%).

### 2.2. Determination of Formulation Adequacy for Skin Application and Protection: In Vitro SPF Assessment, pH, and In Vivo Skin Irritation Sensitization Study

The capacity for skin protection from the harmful effects of UV rays was assessed by determining the developed nanogel’s SPF value, using the classical in vitro spectrophotometric method. The results showed that the developed formulation had a good sun protection factor value, SPF = 34, which proved that this nanogel could be applied for skin protection purposes, namely as a daily application sunscreen. This was expected as the formulation had, in its composition, not only a nanosized inorganic and organic filter mixture (ultrafine TiO_2_ and MBBT) but also GO, which has a reported intrinsic SPF-boosting capacity due to its UV light absorption ability. Additionally, the high antioxidant capacity of this natural compound will lead to an additional skin-protecting effect. Hence, the joint effect of the nanosized UV filters, GO, and other natural-based excipients that are part of the developed NLCs (beeswax, carrot seed oil, and shea butter, with additional antioxidant and anti-inflammatory capacity) lead to the developed formulation having potential synergistic skin protection effects, with broad-spectrum effectiveness [54,55,56].

The pH value of the obtained GO-loaded NLC dispersion and resulting nanogel product were 5.7 and 5.9, respectively. These results are close to the pH range of human skin (4.5 to 6.0). Therefore, the pH value of the developed formulations is suitable for topical application without the risk of leading to local irritation, inflammation, or other types of toxicity [57,58,59].

Nevertheless, the adequacy of topical formulations for skin application should be actively assessed. Hence, the Draize skin irritation test was performed on rats for 72 h in acute testing (Figure 6) and 4 weeks in subacute testing (Figure 7).

The test was performed using topical application of the nanogel, and no skin irritation or sensitization was observed, neither in acute nor subacute testing. Hence, it was concluded that the developed formulation is safe for skin administration. This was achieved by including in the nanogel’s composition authorized UV filters at low concentrations, high safety NLC components with a high percent of natural botanical ingredients, and by not using any toxic organic solvents or any incompatible harmful materials. Considering the well-known skin absorption behaviors of the NLC system, which has a long history in skin drug delivery, the developed nanogel, therefore, proved to be appropriate for topical administration and a possible step towards increasing not only the efficacy but also the safety of sun protection products.

## 3. Materials and Methods

### 3.1. Materials

Carrot seed oil was purchased from Talaye Tabiat Co. (Tehran, Iran). Polyglycerol esters (PGE: E475) were provided by Pars Behbood Asia Co. (Mashhad, Iran). Polyglycerol polyricinoleate (PGPR: E 476) was obtained from Shirin Asal Food Industries Group (Tabriz, Iran). Glycerin was from Azaran Taghtire Sahand Co. (Tabriz, Iran). Caprylic/capric triglyceride (Miglyol 812), organic beeswax, shea butter, polyoxyethylene sorbitan monooleate (Tween 80), methylene bis-benzotriazolyl tetramethylbutylphenol (Nano MBBT), and xanthan gum were supplied by BASF Malaysia Sdn. Bhd. (Shah Alam, Malaysia). The aqueous dispersion of ultrafine TiO_2_ (DIS-AB-10W) was obtained from Pars Hayan Pharmaceutical Co. (Tehran, Iran). Gamma-oryzanol (GO) as a gift was from Tsuno Rice Fine Chemicals Co., Ltd. (Wakayama, Japan). Synperonic PE/F 127 (Poloxamer 407, poly (ethylene glycol)-block-poly (propylene glycol)-block-poly (ethylene glycol)) was from Croda Co. (Nettetal, Germany), and ethanol 96% was purchased from Golriz Co. (Miyandoab, Iran).

### 3.2. Methods

#### 3.2.1. GO-Loaded NLC Preparation

GO-loaded NLCs (GO-NLCs) were prepared by employing hot high-shear homogenization and probe sonication methods. In the case of selected optimized formulation, an aqueous phase, including the surfactant Tween^®^ 80, the vegetable humectant glycerin, and deionized water, and a lipid phase involving the potent natural antioxidant GO, the co-emulsifier PGE, and lipids/emollients/moisturizers carrot seed oil, organic beeswax, shea butter, and Miglyol^®^ 812 were separately heated to a temperature of 75 °C until clear solutions were formed. A pre-emulsion was obtained after mixing the aqueous and oil phases using syringe pump (Syringe Pump SP102, FNM Co. Ltd., Tehran, Iran) under a homogenizer (Heidolph, SilentCrusher M., Nuremberg, Germany). Afterward, the obtained mixture was again homogenized by applying high-shear homogenization at a speed of 21,000 rpm for 30 min. The prepared hot nanoemulsion was then sonicated with a probe sonicator (Up200H, Hielscher, Germany) for 10 min at 80 amplitude, with an on–off cycle of 1 min, followed by cooling to allow the recrystallization of the lipid phase and consequently, obtain the NLCs. The characterization of the obtained GO-NLC dispersions was analyzed in terms of their particle size, polydispersity index (PDI), zeta potential, pH measurement, morphology, physical stability, encapsulation efficiency, and in vitro release studies.

#### 3.2.2. Particle Size and Surface Charge Measurement

The average particle size, PDI, and zeta potential of the prepared GO-NLC dispersions were measured using dynamic light scattering (DLS) technique at 25 °C, using a piece of Zetasizer apparatus (Zetasizer Nano ZS, Malvern Instruments Ltd., Malvern, UK). Prior to actual measurement, NLC dispersion was diluted with double-distilled water. The morphology of the optimized GO-NLC dispersion was obtained using field emission scanning electron microscopy (FE-SEM) on a MIRA3 FE-SEM instrument (TESCAN Co., Cambridge, UK) operated at 15 kV accelerating voltage. Before the examination, the GO-NLCs were dispersed onto the laboratory lamel and dried at 37 °C, then transferred to adhesive-coated aluminum pin stubs, and the stubs were coated with a thin layer of gold. For the transmission electron microscopy (TEM) study, the NLCs were diluted with deionized water, and then the diluted sample was placed onto a carbon-coated copper grid before being blotted off using filter paper. After that, the contrast dye containing uranyl acetate was placed onto the grid, left for 2 min, and blotted off with filter paper. Finally, the grids were loaded onto a specimen holder and then into the device (TEM, 150 Kv, Zeiss-Leo 906, Germany).

#### 3.2.3. Encapsulation Efficiency Determination

A volume of 10 mL of GO-NLC dispersion was centrifuged at 500 rpm for 10 min (Universal Centrifuge 320, PIT, Tehran, Iran). After that, the unloaded GO powder, due to low density (0.32 g/cm^3^), floated on the surface of the dispersion medium and was efficiently separated. Then, 1 mL from the middle of the centrifuged NLC dispersion was added to 5 mL of ethanol and was shaken on a shaker for 30 min, followed by new centrifugation (600 rpm for 15 min) in order to destroy the NLC structure and completely release the GO from inside the nanoparticles and into the ethanol phase. The encapsulated GO value was measured using UV/Vis spectrophotometer (Ultrospec 2000, Pharmacia Biotech, Cambridge, UK) at 322 nm. A GO standard curve was previously prepared in concentrations from 0.1 to 1 mg/mL and measured at the same wavelength. The encapsulation efficiency was calculated via the following equation:Encapsulation efficiency (%) = [Loaded GO/Total GO] × 100

#### 3.2.4. In Vitro Release Study

For the evaluation of in vitro GO release from the developed formulation, a GO-loaded NLC sample of 6 mL was poured into a dialysis bag (molecular weight cut-off 12 KDa, Sigma-Aldrich, St. Louis, MO, USA) and dialyzed against 200 mL of release study medium. The release medium used in this assay was a mixture of 70 % *v*/*v* phosphate buffer saline (PBS) and 30% *v*/*v* of ethanol in Tween^®^ 80, pH 5.5 (Metrohm 827, Herisau, Switzerland) in order to ensure sink conditions were met. The study was undertaken with constant stirring (MR 3001 stirring plate, Heidolph, Germany) at 300 rpm, at 32 °C, for a total duration of 72 h. At predetermined times (1, 2, 4, 6, 8, 12, 24, 48, and 72 h), 1 mL of the release media was reserved and replaced by an equal volume of fresh media. The collected samples were then analyzed, with the amount of released GO being determined using UV spectrophotometry at 322 nm. Furthermore, various dependent kinetic models, such as zero-order, first-order, Higuchi, Peppas, Hixson–Crowell, square root of mass, three seconds’ root of mass, Weibull, linear probability, log—probability, reciprocal powered time, nonconventional order 1, and nonconventional order 2, were used for the analysis of the drug release kinetics. The best-fitting kinetic equation model for the experimental results was selected, the reciprocal powered time (RPT) model, having the highest value of the correlation coefficient (R^2^).

#### 3.2.5. Real-Time and Accelerated Stability Studies

In order to determine the real-time physical stability of the formulated GO-NLCs, particle size, PDI, and zeta potential of the optimized nanosystem, dispersion was measured at room temperature on the day of preparation and then again after a 1 month storage period. To infer the long-term stability of the nanogel, the formulation was subjected to several stress conditions described in the scientific literature, such as high temperatures and elevated relative humidity (RH) [60,61]. In this method, the formulated nanogel was divided into four separate samples, and then these samples were stored in stability chambers under different storage conditions: in a refrigerator at 8 °C; at room temperature (25 °C); in an incubator at 40 °C; or at 40 °C with a 75% RH. These samples were then investigated for changes in different physicochemical parameters, such as changes in color, smell, pH, homogeneity, and phase separation, determined at months 0, 1, 2, 3, and 6.

#### 3.2.6. Topical Nanogel Development

In order to develop a new nano-based sunscreen emulgel product, a low amount of the UV filters ultrafine TiO_2_ and Nano MBBT, in nanosized aqueous dispersion form, were added to the prepared GO-NLCs. This addition was undertaken at room temperature, under nonstop stirring, at a speed of 8000 rpm, using a high-shear homogenizer (Ultraturrax, IKA, Staufen im Breisgau, Germany) for 15 min. This mixture was then converted into the desired nanogel form by adding the natural thickener agent xanthan gum. The prepared finished product was then tested for pH, in vitro SPF value determination, and skin sensitization and irritation potential, as described in the following Section 3.2.7 and Section 3.2.8.

#### 3.2.7. In Vitro SPF Assessment

The SPF value of the prepared topical nanogel was determined using the classical in vitro spectrophotometric method according to the procedure described in ISO 24443:2012, which was carried out in “Specialized Azma Nanosystem Laboratory of Iran” using a Labsphere UV-2000 S device. For this purpose, a sample of 2 mg/cm^2^ of the formulation was applied to a Transpore™ membrane, which mimics natural skin, and UV absorption of the sample was recorded from 290 to 400 nm [62].

#### 3.2.8. Skin Irritation and Sensitization Study

The skin irritation potential of the prepared nanogel formulation was investigated using the Draize test on albino rats [63,64]. In this method, the animals were divided into two groups, each group containing four rats. Hairs were depleted from the back sides of the rats’ bodies with the help of depilatories. One side served as the control group (normal saline), while the other side served as the formulation group (nanogel). The formulation was applied once a day for 72 h in a quantity of 500 mg (0.9% GO, 2% MBBT, 1.8% TiO_2_, 6% liquid oils, and 3% solid lipids) for acute testing, and the same amount was applied for 4 weeks in subacute testing. Skin irritation potential of the developed formulations was then determined using observations of occurrence of any skin sensitivity and reactions, such as redness, edema, and rash. Then, the skin irritation potential of the developed nanogel was graded using the following classifications: 0 no reaction; 1 slight patchy erythema; 2 moderate patchy erythema; 3 moderate erythema; and 4 severe erythema, with or without edema (ethical code: IR.TBZMED. REC.1399.509).

## 4. Conclusions

A new broad-spectrum nanogel product for skin protection was successfully obtained by adding the latest innovative EU-authorized UV filters (ultrafine TiO_2_ and Nano MBBT) in nanosized aqueous dispersion form to natural-based GO-loaded NLCs, followed by the use of a natural polymeric thickener agent to obtain a final nanogel form. The main purpose of this study was to create an innovative and useful approach for skin protection, especially against harmful UV rays, based on a new generation of nanodrug delivery systems by the reduction of synthetic UV filter use and replacement with naturally sourced ingredients in order to limit the side effects and improve efficiency, with a suitable texture and sensorial experience on the consumer’s skin. The developed NLCs showed good physical stability, with electronegative zeta potential value (−34.9 mV), small and adequate particle size (146.5 nm), narrow size distribution (PDI 0.204), homogenous spherical structure, high encapsulation efficiency, and a controlled release profile. Additionally, the prepared nanogel indicated appropriate UV shielding capacity (SPF = 34) due to the presence of an inorganic and organic UV filter mixture, the potent plant antioxidant GO, and natural oil–lipid components in the NLC formulation base, with synergistic effects. The developed formulation also had good stability in long-term storage of up to 6 months (accelerated stability studies) and led to no skin irritation or sensitization, making it suitable for topical application. Hence, the prepared nanogel proved to have skin photoprotection properties, being a multifunctional product that could have antiaging efficacy due to delaying the process of photoaging and potentially decreasing the probability of the occurrence of skin diseases, such as cancer.

## Figures and Tables

**Figure 1 pharmaceuticals-16-00670-f001:**
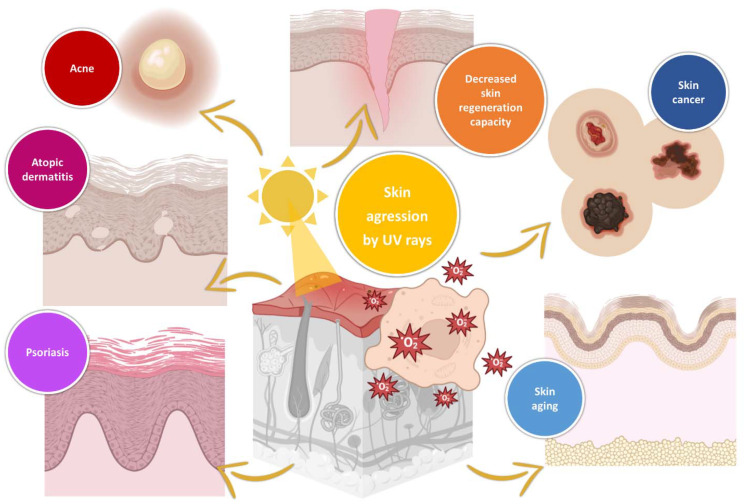
Skin diseases that will have an increased propensity for development due to UV radiation exposure (produced using BioRender).

**Figure 2 pharmaceuticals-16-00670-f002:**
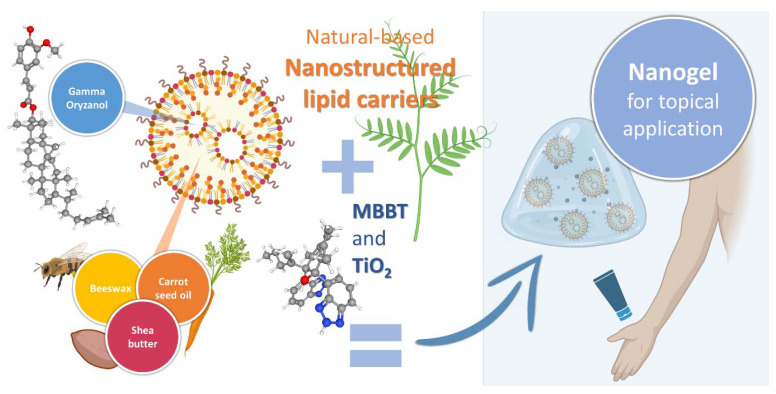
General composition of the developed topical nanogel containing GO-loaded NLCs and nanosized UV filters TiO_2_ and MBBT (produced using BioRender).

**Figure 3 pharmaceuticals-16-00670-f003:**
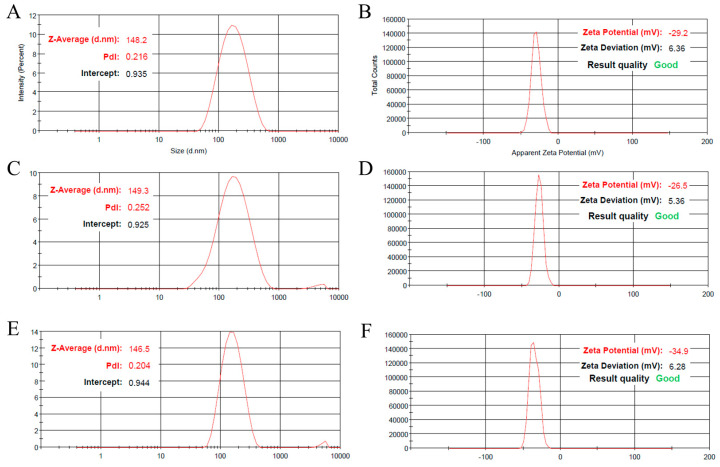
Particle size distribution and zeta potential values of the optimized blank nanostructured lipid carriers immediately after preparation (**A**,**B**) and after 1 month of storage (**C**,**D**), and gamma-oryzanol-loaded NLCs (**E**,**F**).

**Figure 4 pharmaceuticals-16-00670-f004:**
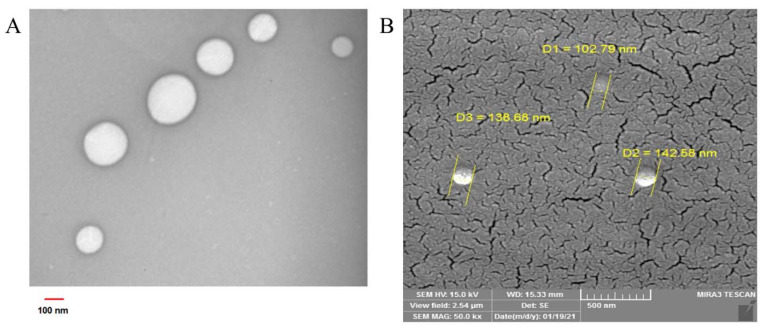
Scanning electron microscopy (**A**) and transmission electron microscopy (**B**) photographs of the gamma-oryzanol-loaded nanostructured lipid carrier formulation.

**Figure 5 pharmaceuticals-16-00670-f005:**
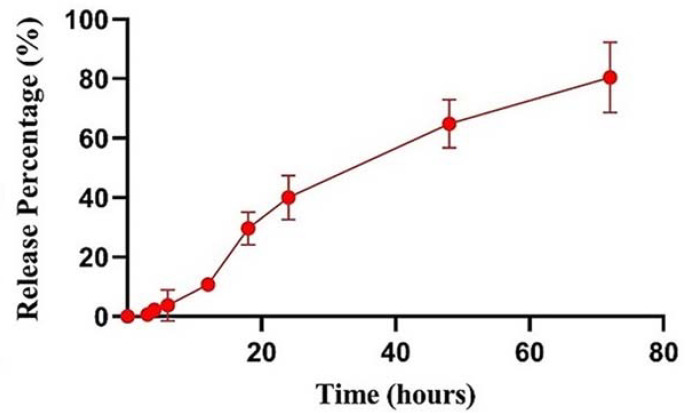
In vitro drug release profile of gamma-oryzanol-loaded nanostructured lipid carrier dispersion.

**Figure 6 pharmaceuticals-16-00670-f006:**
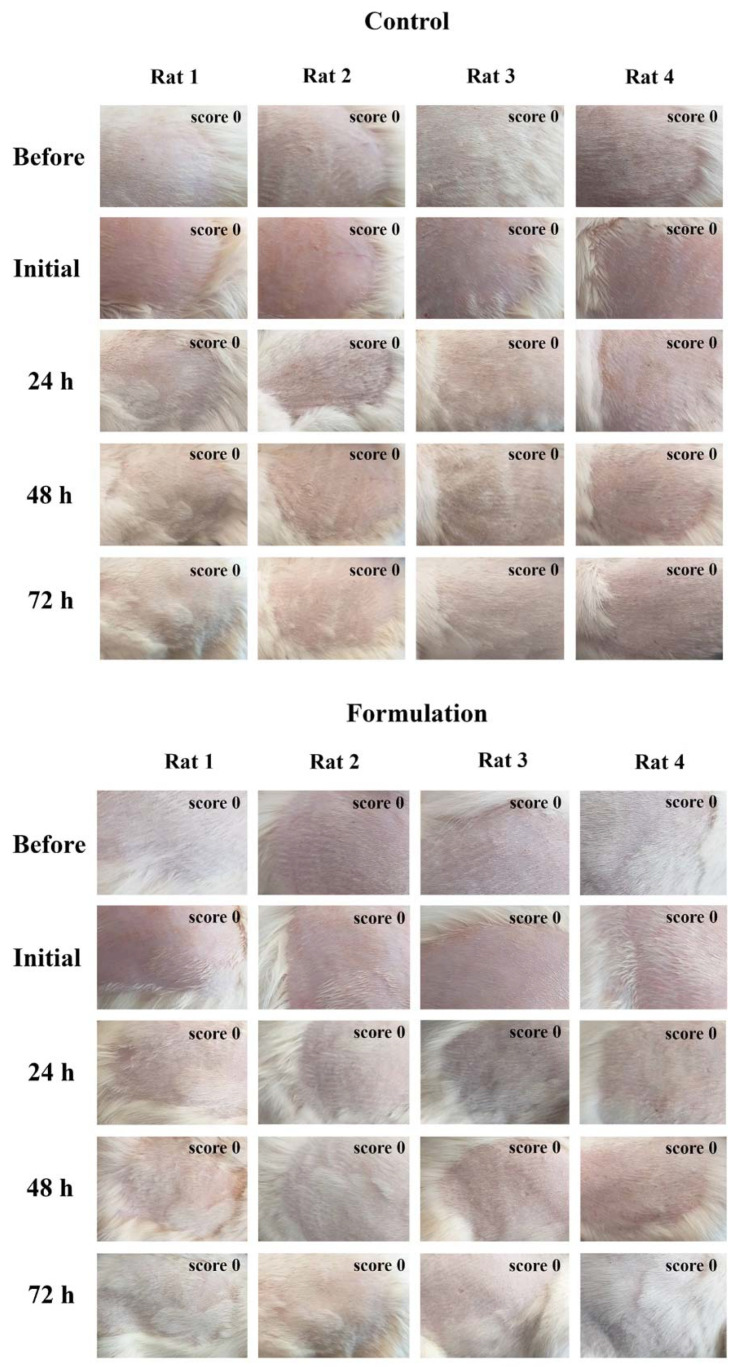
Photographs of Draize skin irritation test in acute testing carried out on albino rats; in the control group, normal saline was applied, and in the formulation group, the developed nanogel was applied; score corresponds to minimum erythemal score.

**Figure 7 pharmaceuticals-16-00670-f007:**
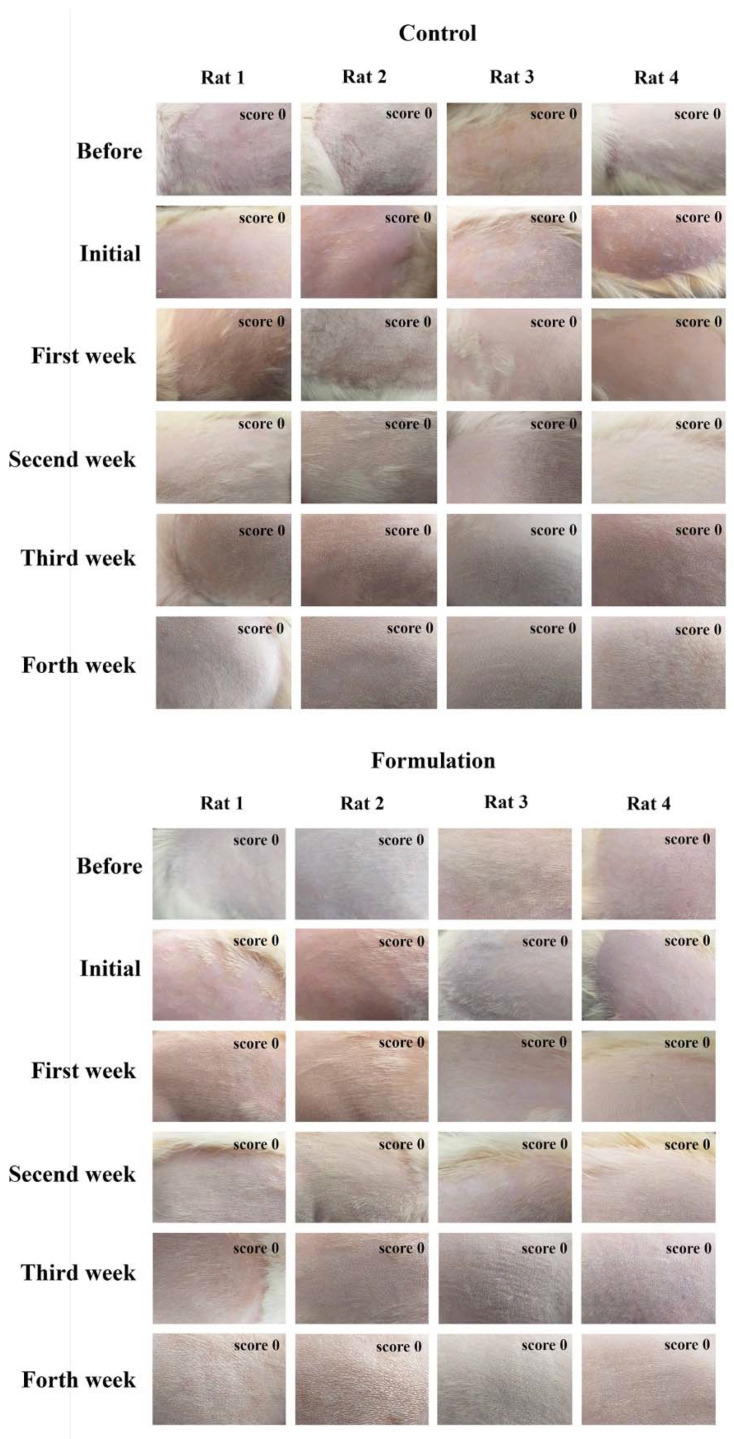
Photographs of Draize skin irritation test in subacute testing carried out on albino rats; in the control group, normal saline was applied, and in the formulation group, the developed nanogel was applied; score corresponds to minimum erythemal score.

**Table 1 pharmaceuticals-16-00670-t001:** Tested formulation composition variations for the design of the gamma-oryzanol-loaded nanostructured lipid carriers, with resulting particle size and PDI values.

NLC Formulation Code	Coemulsifier (Lipid Phase)		Surfactant (Aqueous Phase)		Oil–Lipid	Results	
						Size ^a^	PDI ^b^
	Type	Percent	Type	Percent	Percent	(nm)	
F1	-	-	Poloxamer 407	3	7.6	280	0.29
F2	-	-		2.5	7.6	195	0.28
F3	-	-		2	7.6	195	0.17
F4	PGPR ^c^	1		2.5	7.6	286	0.39
F5		1		2	7.6	190	0.15
F6		1		1	7.6	380	0.34
F7		1.1		2.3	8.6	181	0.28
F8		1.2		2.4	9	189	0.22
F9		1	Tween^®^ 80	2	7.6	241	0.26
F10	PGE ^d^	1		2	7.6	209	0.37
F11		1.5		2	7.6	189	0.34
F12		1		2.5	7.6	208	0.41
F13		1		2.5	8	215	0.46
F14		1.6		2.1	8	151	0.28
F15		1.8		2.4	9	148	0.21
F15 after 1 month storage		1.8		2.4	9	149	0.25
F16 (loaded with GO)		1.8		2.4	9	146	0.20

^a^ Based on z-average distribution; ^b^ polydispersity index; ^c^ polyglycerol polyricinoleate; ^d^ polyglycerol esters; blank NLC formulations: from F1 to F15; gamma-oryzanol (GO)-loaded NLC formulation: F16.

## Data Availability

Not applicable.

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
