# Peer review of "Nanogel Containing Gamma-Oryzanol-Loaded Nanostructured Lipid Carriers and TiO2/MBBT: A Synergistic Nanotechnological Approach of Potent Natural Antioxidants and Nanosized UV Filters for Skin Protection"

_pharmaceuticals, 2023, doi:10.3390/ph16050670_

Round 1

Reviewer 1 Report

1- On what basis the authors choose the different factors for formulations. Did they use design expert. Please explain more.

2- Why did not the authors compare the free drug to the loaded one in the release study.

3- DId the authors do any kinetics for the release study.

4- Please explain more the methodology for calculating SPF.

5- The authors mentioned one time that the scoring system for skin irritation is A, B, C, etc while at the figures it is numerical. So it should be unified.

Author Response

We thank the reviewer for their insightful and knowledgeable comments and suggestions, and have sought to address each and every aspect carefully (alterations in the manuscript marked by track changes). A point-by-point answer is given bellow.

  1. On what basis the authors choose the different factors for formulations. Did they use design expert. Please explain more.

Response: We thank the reviewer for their comment. Considering the well-known role of the type and ratio of liquid oils, solid lipids and surfactants in the structure of the nanostructured lipid carriers (NLCs), and also due to the experience of our research group in developing various formulations based on NLCs, instead of using computational/statistical tools for the experimental design, our main focus was on developing an effective nanostructured lipid-based sunscreen formulation partly replacing UV filters by including and maximizing the amount of the potent plant antioxidant GO in the formulations. Additionally, we chose to use natural oil-lipids due to their proved skin protection effects and environmentally friendly nature, and these components allowed to use a smaller amount of UV filters in the formulation, having synergistic effects on SPF value, and leading to a suitable texture in terms of consistency and sensorial experience on the consumer's skin. To increase reader understandability a more complete and thorough explanation has been added to section 3.1.

  1. Why did not the authors compare the free drug to the loaded one in the release study?

Response: We thank the reviewer for their question. In fact, sometimes in in vitro drug release studies the free drug is compared to the developed formulations. Nevertheless, as we had established the sink conditions for the drug release study using Tween 80 and ethanol, it was highly expected that the free drug would show 100% of drug release within the first time points, and hence we concluded that this evaluation would not add relevant information to the study, hence not performing it.

  1. Did the authors do any kinetics for the release study?

Response: We thank the reviewer for their suggestion. Drug release kinetics are in fact a very relevant parameter, and hence this information has been added to section 2.2.4. and section 3.1. In order to determine the suitable drug release kinetic model, various dependent kinetic models such as Zero-order, First-order, Higuchi, Peppas, Hixson-Crowell, Square root of mass, Three seconds root of Mass, Weibull, Linear probability, Log – probability, Reciprocal powered time, Non conventional order 1 and Non conventional order 2 were used. Reciprocal Powered Time model had the best fitting kinetic equation model for the experimental results, having a maximum value of RSQ = 0.994 and minimum amount of MPE = 11.78.

  1. Please explain more the methodology for calculating SPF.

Response: Thank you for your suggestion. The SPF value measurement was done in accordance with protocols described in the ISO 24443:2012 method which was carried out in “Specialized Azma Nanosystem Laboratory of Iran” by Labsphere UV-2000 S device. This information was added to section 2.2.7 of the main text. The related reference was added in the revised article, instead of previous ref. [49].

  1. The authors mentioned one time that the scoring system for skin irritation is A, B, C, etc while at the figures it is numerical. So it should be unified.

Response: We thank the reviewer for noticing, and as suggested, the mentioned inconsistency was resolved in section 2.2.8 of the main text.

Reviewer 2 Report

The experimental investigation is well organised presented, however to make the manuscript more interesting to reader, the following corrections are suggested.

1. Line no 249-251. Suggested to remove does not provide significant informations

2. Suggested to add ethic number for animal study in section 2.2.8

3. Suggested to calculate the quantity of active ingredient in equivalent to 500 mg of nano gel applied over animal to perform acute testing

4. Please report the value of antioxidant to support the statement "on Line 370-321" either by performing the experiment or from previous report

5. Supplement the data with results and images obtained from stability study. (example to present supplement the data for UV-protective formulation:https://doi.org/10.3390/gels8100639)

6. Suggested to rewrite the conclusions, it more or less looking like summarise results.

7. There are some typing error and grammatical error reflected in the attached file, suggested to look on that.

Thanks and Good Luck

Author Response

We thank the reviewer for their insightful and knowledgeable comments and suggestions, and have sought to address each and every aspect carefully (alterations in the manuscript marked by track changes). A point-by-point answer is given bellow.

  1. Line no 249-251. Suggested to remove does not provide.

Response: We thank the reviewer for the suggestion, and hence the correction was made, in section 2.2.5.

  1. Suggested to add ethic number for animal study in section 2.2.8.

Response: Thank you for the suggestion. According to the reviewers' comment, the ethics number for the animal study was moved from the “Funding” section to the 2.2.8. section.

  1. Suggested to calculate the quantity of active ingredient in equivalent to 500 mg of nano gel applied over animal to perform acute testing.

Response: We thank the reviewer for their suggestion. The quantity of active ingredients equivalent to 500 mg of nano gel was calculated, being 0.9% GO, 2% MBBT, 1.8 % TiO2, 6% liquid oils and 3% solid lipids. This information was added to section 2.2.8.

  1. Please report the value of antioxidant to support the statement "on Line 370-321" either by performing the experiment or from previous report.

Response: Thank you for the suggestion. The high antioxidant capacity of GO was already present in lines 116 to 117. This information has now been completed, and a new reference was also added (now lines 122 to 123).

  1. Supplement the data with results and images obtained from stability study. (example to present supplement the data for UV-protective formulation: https://doi.org/10.3390/gels8100639).

Response: We thank the reviewer for their suggestion. As mentioned in section 2.2.5, the stability study was done in two ways: the first one was determination particle size, PDI and zeta potential of the prepared GO-NLC at room temperature, first on the day of preparation and then again after a 1 month storage period. The second one was done via investigation of the organoleptic properties in physicochemical parameters of the obtained nanogel in the case of changing in color, smell, pH, phase separation and homogeneity during 6 months. Since GO-NLC were transformed into a nanogel, and this was the final formulation, presenting supplementary data including particle size, PDI and zeta potential variation GO-NLC during a 6 month storage would not be representative of the final formulation.

  1. Suggested to rewrite the conclusions, it more or less looking like summarise results.

Response: Thank you for the suggestion. The conclusion section has now been modified to become more comprehensive and to summarize the results.

  1. There are some typing error and grammatical error reflected in the attached file, suggested to look on that.

Response: Thank you for the suggestion. Language was thoroughly checked and all the identified grammatical and/or typographical errors were corrected.

Reviewer 3 Report

In the present work, Badalkhani et al. discusses the development of a topical xanthan gum nanogel containing Gamma Oryzanol-loaded nanostructured lipid carriers (NLC) and nanosized UV filters TiO2 and methylene bis-benzotriazolyl tetramethylbutylphenol (MBBT) for skin protection. The combination of natural antioxidants and UV filters delivered through nanotechnology is expected to have multifunctional skin beneficial properties. The article provides a clear and concise introduction to the topic of skin protection and the use of nanotechnology in this field. The methodology used in the study is well-described, with detailed information on the materials and methods used for creating the nanogel-containing Gamma Oryzanol-loaded NLC and UV filters. However, some issues must be addressed before acceptance for publlication.

1. It would be essential to include more information on how this approach compares to other existing methods for skin protection, such as traditional sunscreens or other natural remedies, as well as any limitations or challenges that may arise in its implementation (e.g., cost, availability).

2. Some sections of the article could benefit from further elaboration to provide more clarity on certain aspects of the research, such as how exactly the nanogel works to deliver antioxidants and UV filters (TiO2) to the skin.

3. There may be potential risks or side effects associated with using this nanotechnological approach for skin protection, it would be encouraged to provide more detailed information on these risks and how they can be mitigated.

Minor editing of English language required

Author Response

We thank the reviewer for their insightful and knowledgeable comments and suggestions, and have sought to address each and every aspect carefully (alterations in the manuscript marked by track changes). A point-by-point answer is given bellow.

  1. It would be essential to include more information on how this approach compares to other existing methods for skin protection, such as traditional sunscreens or other natural remedies, as well as any limitations or challenges that may arise in its implementation (e.g., cost, availability).

Response: We thank the reviewer for their comment. Among the concerns about some current sunscreen products in the market is the use of some dangerous chemical filters, such as oxybenzone, avobenzone, octocrylene, and octyl cinnamate, with the possibility of creating toxicity and sensitivity in long-term use of the product, and causing defects in the normal functioning of tissues and internal organs of the body. Therefore, in designing the formulation of this sunscreen product, an efficient and safe solution was applied to improve the common commercial sunscreen formulations, by developing a new biocompatible and biodegradable nanoparticle carrier system of nanostructured lipid carriers (NLCs), with the aim of limiting side effects by replacing the above-mentioned chemical UV filters by maximum amount of potent plant antioxidant GO and natural oil-lipids components in combination with less amount of the latest innovative nanostructured UV filters, coupled with synergistic effects which can able to provide a broad-spectrum sun protection and preventing premature aging caused by the radiation of dangerous waves of sunlight. We believe it is a step towards increasing safety of sun protection products. This information has now been complemented at the end of section 3.2.

  1. Some sections of the article could benefit from further elaboration to provide more clarity on certain aspects of the research, such as how exactly the nanogel works to deliver antioxidants and UV filters (TiO2) to the skin.

Response: Thank you for the suggestion. The colloidal character of NLCs lead to represent intelligent encapsulation systems with especially advantageous to design transdermal drug delivery. Also, this system has a long history in topical use with obvious absorption mechanisms which were added in the main text in line 427-431.

  1. There may be potential risks or side effects associated with using this nanotechnological approach for skin protection, it would be encouraged to provide more detailed information on these risks and how they can be mitigated.

Response: We thank the reviewer for their suggestions. NLC systems are attracting major attention as novel colloidal drug carriers with biodegradable and safe for human use due to their non-toxic ingredients. It has a long history in many desired cosmetic formulations with good perspectives to be marketed very successfully, reason for this is they were developed considering industrial needs, e.g. scale up, qualification and validation, simple technology, low cost, tolerability etc. First products were Nanorepair and Nanovital by Dr. Rimpler GmbH in Germany in 2005, followed in 2006 by 3 products in the line IOPE by AmorePacifi c in South Korea. Therefore, nowadays, many cosmetic products containing lipid nanocarriers are available in the market which are free of any risks or side effects for skin protection. This information has now been added/complemented, from lines 96 to 102, with the related references (12-15).

Round 2

Reviewer 3 Report

The authors have adressed my issues, thus I recommend that their work can be accepted in its current state.